# Development of Prostate Cancer Organoid Culture Models in Basic Medicine and Translational Research

**DOI:** 10.3390/cancers12040777

**Published:** 2020-03-25

**Authors:** Mohamed Elbadawy, Amira Abugomaa, Hideyuki Yamawaki, Tatsuya Usui, Kazuaki Sasaki

**Affiliations:** 1Laboratory of Veterinary Pharmacology, Department of Veterinary Medicine, Faculty of Agriculture, Tokyo University of Agriculture and Technology, 3-5-8 Saiwai-cho, Fuchu, Tokyo 183-8509, Japan; mohamed.elbadawy@fvtm.bu.edu.eg (M.E.); s193249s@st.go.tuat.ac.jp (A.A.); skazuaki@cc.tuat.ac.jp (K.S.); 2Department of Pharmacology, Faculty of Veterinary Medicine, Benha University, Moshtohor, Toukh 13736, Elqaliobiya, Egypt; 3Faculty of Veterinary Medicine, Mansoura University, Mansoura 35516, Dakahliya, Egypt; 4Laboratory of Veterinary Pharmacology, School of Veterinary Medicine, Kitasato University, Towada, Aomori 034-8628, Japan; yamawaki@vmas.kitasato-u.ac.jp

**Keywords:** organoid, prostate cancer, precision medicine, translational research, drug resistance

## Abstract

Prostate cancer (PC) is the most prevalent cancer in men and the second main cause of cancer-related death in Western society. The lack of proper PC models that recapitulate the molecular and genomic landscape of clinical disease has hampered progress toward translational research to understand the disease initiation, progression, and therapeutic responses in each patient. Although several models have been developed, they hardly emulated the complicated PC microenvironment. Precision medicine is an emerging approach predicting appropriate therapies for individual cancer patients by means of various analyses of individual genomic profiling and targeting specific cancer pathways. In PC, precision medicine also has the potential to impose changes in clinical practices. Here, we describe the various PC models with special focus on PC organoids and their values in basic medicine, personalized therapy, and translational researches in vitro and in vivo, which could help to achieve the full transformative power of cancer precision medicine.

## 1. Introduction

Prostate cancer (PC) is one of the most prevalent cancers worldwide and the second common cause of cancer-related deaths among American men [1]. It is apart from hereditary cases with a peak incidence in aged men (75 years~) and the epidemiology remains unclear [2]. Human PC mainly exhibits low grade with slow progression [3] and can be easily diagnosed by measuring the plasma concentration of prostate-specific antigen (PSA) protein [4]. Although the overall death rate due to PC is gradually decreasing due to the improvements in early diagnosis and prevention, the late diagnosis of PC leads to advanced and metastatic disease in which the survival of patients shortens. In this case, the androgen deprivation therapy plus gonadal repression and/or chemotherapy is usually performed [5]. Despite initial responses, castration resistance ultimately ensues. Therefore, the development of new drugs with higher specificity and decreased toxicity is of utmost need in the severe PC disease. 

The etiological origin of PC remains unclear due to the heterogeneity of the gland tissues [6]. Histologically, the prostate gland is composed of basal, luminal, and neuroendocrine cells embedded in a fibro-muscular stroma [7]. The basal layer (contains stem, transit-amplifying, and basal committed cells), constructs up to 40% of the total epithelium and expresses p63, cytokeratin (CK) 5, and estrogen receptor (ER) β. Due to the lack of androgen receptor (AR), they are androgen-independent. The luminal secretory cell layer accounts for the rest of epithelium and expresses CK 8/18, AR, and ER β [8]. They are, therefore, dependent on androgen for their growth and survival [6,9]. The stroma of human prostate expresses mainly ERα, while the neuroendocrine cells express chromogranin A and synaptophysin and have unknown functions.

It has been reported that receptors for androgen and estradiol play important roles in mediating the hormonal control of prostate cell stemness, transformation, growth, proliferation, invasiveness, and metastasis [8,10,11]. The androgen, through AR, enters the prostate cell nuclei and activates gene transcription [12]. Also, the ligand-bound AR acts in target cells at the non-transcriptional level [13], triggering the fast and temporal stimulation of Src tyrosine kinase which rises the active form of small-GTP binding proteins (Ras and Rac 1) and activates mitogen-activated protein kinases as well as focal adhesion kinase in PC cells [14]. Moreover, androgens also promote cell proliferation through the activation of PI3-K/Akt pathways [15]. Additionally, the androgen-triggered AR/filamin A complex is required for the motility, migration, and adhesion of NIH3T3, HT1080, and neuronal PC12 cells [16].

Estrogens, with androgens, trigger the hyperplasia and transformation of prostate as well as PC progression [17,18]. The high ER expression is associated with PC risk. Estrogens, through ERα and β, stimulate the signal transduction pathways including Src, adenylyl cyclase, MAPKs, and PI3-K, or mediate a rapid increase in intracellular calcium levels [14]. Migliaccio et al., reported that a rapid estradiol action mediated by ER β leads to proliferation and cell cycle progression in LNCaP cells [19]. Also, estrogens via ER play a role in epithelial versus mesenchymal transition of prostate cancer cells [10]. Therefore, the selective targeting of AR and ER (α or β) may be an attractive mean to restrict the growth and proliferation of PC cells.

In PC, the predisposing factors leading to a disruption of the epithelial lineage and basement membrane, are followed by a drift of cell population numbers with tumor growth via heterotypic signaling [20,21]. Although clarifying the cell origin of PC is important for establishing the new therapeutic strategy, it is the subject of considerable discussion. 

Establishing experimental model systems for PC that precisely mimic both the genetic divergence and lineage specificity of cancer is critical to perceive the function of cancer-related genetic changes in tumorigenesis, tumor maintenance, and therapeutic sensitivity or resistance.

The 3D organoid culture systems have been shown to be valuable in clarifying the biology of various types of cancer diseases [22,23]. In PC, it is also useful in identifying tumor-initiating cells from an epithelial cell lineage. The patient-derived PC organoids can provide preclinical models to perceive the precision medicine by drug screening of individual patient samples [24]. 

In the present review, we discuss the potentials of PC organoids to identify the cell of origin, to study PC microenvironment, to screen genetic mutations, and to check drug sensitivity in each PC patient.

## 2. Treatment Challenges Against PC

Several kinds of treatments were used in PC, some of which are associated with drawbacks (Figure 1). The low grade of local PC is usually treated with surgical resection (radical prostatectomy) and/or radiotherapy (brachytherapy or external beam radiotherapy) [25], followed by routine monitoring of PSA level in the blood to predict the recurrence [26]. However, prostatectomy is usually followed by erectile dysfunction and urinary incontinence [27,28]. Radiotherapy is also associated with more bowel and rectal disturbances [29]. 

Since most PC patients are androgen-driven adenocarcinoma, androgen-deprivation therapy (ADT) is the main treatment for patients with locally advanced, recurrent, and metastatic disease [30] with or without the addition of chemotherapy or the potent androgen synthesis inhibitor, abiraterone acetate [31,32]. 

Although ADT is effective at first, the long-lived patients finally develop a metastatic castration-resistant PC (mCRPC), the more advanced form of the disease with a 2–3 years survival rate [33]. Antiestrogen-based therapy also has values in the treatment of PC [34]. Although immunotherapy has shown fundamental aids for several types of cancer, it only presented a marginal benefit for mCRPC, probably due to its immunosuppressive tumor microenvironment [35]. Although immunotherapy based on the blockade of immune checkpoint inhibitors plays a role in the treatment of most advanced cancers [36], PC is still in its infancy [37].

## 3. Experimental Models to Study PC

Establishing experimental model systems either in vitro or in vivo that precisely mimic the genetic divergence, lineage specificity, microenvironment, and pathophysiology of the tumor is critical to perceive the function of cancer-related genetic changes in tumorigenesis, tumor maintenance, and therapeutic sensitivity or resistance in PC disease. As shown in Figure 2, the experimental models have been established in the previous studies.

### 3.1. PC Cell Lines

Cell lines have been used for realizing the function of genetic and molecular changes in cancer such as the discovery of genomic biomarkers of anti-cancer drug sensitivity [38,39,40]. In PC, the scarcity of classical cell line models is mainly due to the difficulty in propagating PC cells in vitro for a long period, because most PC cells do not grow in a traditional culture medium. Despite the high prevalence of PC among men worldwide, only a few PC cell lines have been established via many trials by multiple researchers. As shown in Table 1, these cell lines were established from original tissues of PC, metastasized PC, and PC xenograft models from the primary tumor or the metastasized ones. Examples of these cell lines are LNCaP [41,42], PC-3 [43,44], DU145 [45,46],VCaP [47,48], NCI-H660 [49], PC346 [50], 22Rv1 [51], KUCaP [52], MDA PCa 2a and MDA PCa 2b [53,54]. Among them, DU145, PC-3, and LNCaP are the most widely used ones in PC research [55,56].

Additionally, several sublines have been established from the classical PC cell lines, particularly the LNCaP one. While the original LNCaP cells are sensitive to androgen, their sublines, LNCaP-abl and LNCaP-LTAD, generated by removing androgen from the culture medium, are androgen-resistant [57,58]. Moreover, various resistant PC sublines have been generated by incubating the original cell lines with anti-androgen or chemotherapeutic agents (Table 2). 

Although there are the ease of use, reproducibility, and cost-effectiveness, the major limitations of PC cell lines are genetic shift or acquisition of mutations due to long-term culture. Recent papers showed important genetic alterations in PC patients, which might regulate prostate tumorigenesis [59,60], many of which (e.g., CHD1 loss, Forkhead box protein (FOX)A1 mutation, Speckle-type POZ protein (SPOP) mutation, etc.) are not demonstrated in PC cell lines [61]. Moreover, the lack of relevant clinical data, loss of tumor heterogeneity, and 2D cell lines do not accurately recapitulate the 3D tumor microenvironment, which is especially important in cancer research. While PC cell lines have helped progress the PC research field, they do not reflect the clinical spectrum of PC [62].

To overcome these limitations of the PC cell lines, various *in vivo* models of tumor propagation have been developed by researchers, including patient-derived xenograft (PDX) models, genetically engineered mouse (GEM) models, and 3D culture of patient-derived PC cells (organoid or sphere culture).

**Table 1 cancers-12-00777-t001:** Original PC cell lines and their pathological type, origin, prostate-specific antigen (PSA) reactivity, and expression level of androgen receptor (AR) and estrogen receptor (ER).

Cell Line	Origin	Type	PSA	AR	ER-α, β	Reference
LNCaP	Lymph node metastasis	Adenocarcinoma	+	+	+ ER-β	[41,42]
PC-3	Bone metastasis	Adenocarcinoma	-	-	+ ER-α, ER-β	[43,44]
DU145	Brain metastasis	Adenocarcinoma	-	-	+ ER-β	[45,46]
VCap	Xenograft from metastasis	Adenocarcinoma	+	+	+ ER-β	[47,48]
NCI-H660	Metastatic extrapulmonary small cell carcinoma arising from the PC	Adenocarcinoma	-	-	-	[49]
PC346	Primary tumor xenograft	Adenocarcinoma	+	±	-	[50]
22Rv1 (CWR22Rv1)	CRPC derivative of CWR22 xenograft	Adenocarcinoma	+	+	+ ER-α, ER-β	[51]
CWR22Pc	CWR22 xenograft	Adenocarcinoma	+	+	+ ER-β	[63,64]
MDA PCa 2a & MDA PCa 2b	Bone metastasis	Adenocarcinoma	+	+	+ ER-β	[53,54]
KUCaP	Xenograft from liver metastasis	Adenocarcinoma	+	+	-	[52]

CRPC: Castration-resistant prostate cancer; AR: androgen receptor-dependent.

**Table 2 cancers-12-00777-t002:** Sublines derived from PC cell lines and their resistant profile.

Subline	Original Cell Line	Resistance	Established Method	Reference
LNCaP-abl	LNCaP	Castration	Culture in ADM	[57]
LNCaP-LTAD	LNCaP	Castration	Culture in ADM	[58]
LNCaP-BicR	LNCaP	Anti-androgens	Culture with flutamide	[65]
PC346Flu1/2	PC346	Anti-androgens	Culture in ADM with flutamide	[66]
PC-3R	PC-3	Chemotherapy	Docetaxel	[67]
DU145-TxR	DU145	Chemotherapy	Culture with paclitaxel	[68]
DU145R	DU145	Chemotherapy	Culture with docetaxel	[67]
PC-3CR	PC-3	Chemotherapy	Culture with cabazitaxel	[69]
PC-3 D12	PC-3	Chemotherapy	Culture with docetaxel	[70]

AAR: anti-androgen agent resistant; ADM: androgen-depleted medium.

### 3.2. Xenograft Mouse Models

To date, preclinical PC mouse models continue to be an important gadget to enhance our perception of PC development, proliferation, and metastatic behavior. The incidence of spontaneous PC in mice is rare [71]. Therefore, several tumor-xenografted or genetically engineered PC mouse models have been developed [72] (Figure 1). Among them, xenograft models have been emerged and generated through heterotopic or orthotopic [73] implantation of human tumor tissues, primary cell cultures, or cell lines [74], in nude mice [75], SCID [76], NOD-SCID [77], NOG/NSG [78], or RAG [79]. 

Compared with cell lines, patient-derived xenografts (PDXs) may more properly recapitulate the molecular divergence and cellular heterogeneity of tumors of patients [80,81]. The PDXs of PC were easily developed, characterized [82,83], and shown to be beneficial in anti-cancer drugs screening for efficacy and toxicity [84,85]. Xenografting of patient-derived PC cells was improved by using chimeric grafts with neonatal mouse mesenchyme [86] and by establishing highly immunodeficient mice such as NOG (NOD/Scid/IL2Rγ^null^), NSG (NOD/Scid/IL2Rγ^null^), and NOJ (NOD/Scid/Jak3^null^) mice [87]. 

Several PDX models are established by directly xenografting patient-derived tissues into immunodeficient mice. The first of which has been demonstrated by Gittes in 1980 in athymic nude mice and maintained many of the histological characteristics of human PC [88]. Later, numerous models were established, as reviewed in [89]. For example, the Rotterdam PC xenograft models were generated from primary prostatectomy specimens, transurethral resection specimens, and metastatic lesions [50,75,90], and the established xenografts retained the histological structure of their original patient tumors. Another model is the LuCaP PDX model which was established from primary PC tumors or PC metastases [84,91] and showed the main genomic and phenotypic characters of their original tumors, including Phosphatase and tensin homolog (PTEN) deletion, AR amplification, TP53 deletion and mutation, Transmembrane protease, serine (TMPRSS)-Erythroblast transformation-specific-related gene (ERG) rearrangement, RB1 loss, SPOP mutation, and BRCA2 loss. 

Despite the beneficial outcomes from using mice in xenograft studies, the heterogeneity of mouse tissues regarding the cancer stem cell (CSC) niche and stromal compartment of the prostate [92] compared with human and immunodeficiency state are major limitations [81]. Additionally, different murine physiology and response to therapeutic agents are also other potential limitations. Moreover, a drug screening using xenograft model mice usually takes several months and costs a lot of money. Therefore, it is not amenable to high-throughput screening [93]. 

### 3.3. Organoid Culture of Patient-Derived PC Cells

Because of the limitations of cell lines and PDXs, 3D cell culture systems (known as organoids) are getting great attention as patient-derived cancer models. Organoids are mini organ-like structures with important organ features. It is frequently used as an intermediate model between in vitro cancer cell lines and PDXs. Organoids can efficiently and closely recapitulate the in vivo microenvironment as well as molecular and genetic signature of tissues or organs of origin and are capable of self-renewal and self-organization [94,95,96,97]. It also could provide the benefit for cancer-related studies, disease modeling, drug discovery, and personalized therapy [95,98,99,100,101,102,103]. The microenvironment, particularly the extracellular matrix, in which the organoids are grown, strongly influences their cellular behavior [104].

In addition to the primary 3D organoid culture system, numerous 3D culture model systems have been established to simulate the in vitro structural connections between epithelial cells and stroma and to model organ development and function [105]. These systems are derived from cell lines, primary tissues, embryonic stem cells, and induced pluripotent stem cells (iPS cells) [105] (Figure 2). 

#### 3.3.1. PC Organoids for Screening of Gene Mutations

The incidence of genomic instability is one of the hallmarks of cancer [106,107]. Consequently, cancer cells typically contain numerous mutations, which widely differ among and within different types of cancers [108]. Interestingly, only a few of these mutations (driver mutations), drive the disease progression [107]. Hence, organoids can maintain the genetic and phenotypic landscape of the original tumor and are more suited for in vitro manipulations. Therefore, they were efficiently used to identify the driver mutations of the original tumors [109]. Additionally, organoids can be genetically handled using CRISPR/Cas9 and shRNA systems rendering the organoid culture an attractive platform for rapid examination of the impact of genotypes and mutational signature on pharmacological responses [110]. 

The organoids from the primary advanced or castration-resistant PC tissues or PDXs were successfully shown to recapitulate several genetic mutations of the disease (Figure 3). Next-generation sequencing of the exome and genome of PC has recognized several genetic mutations. Among them, SPOP is one of the most frequently mutated genes in primary PC, suggesting that SPOP may be a potential driver of PC [111]. In mouse PC organoids, mutation of the SPOP gene led to increased proliferation and a transcriptional signature consistent with human PC.

Recently, normal human and mouse prostate organoids have been developed, which can be cultured indefinitely without immortalization and can recapitulate the normal prostate glandular structure [110]. This system has been optimized for human advanced and metastatic PC specimens and 7 new organoids lines have been successfully established (six from biopsy samples and one from circulating tumor cells). After 3 months of in vitro culture, this system clearly demonstrated the previously identified frequent genetic mutations shown in advanced primary PC including *PTEN* loss, *SPOP* mutation, *TMPRSS2-ERG* interstitial deletion, as well as mCRPC involving Phosphoinositide-3-kinase regulatory subunit 1 (*PIK3R1), TP53, FOXA1,* and several mutations in chromatin modifier [112]. In another study, PC organoids generated from PDXs of mCRPC successfully recapitulated and conserved the genetic and phenotypic heterogeneity such as copy number variations including gains at chromosome 8q and losses at 8p and 13q [113]. Moreover, gain of R-spondin (RSPO)2, loss of PTEN, and a somatic coding mutation in Low-density lipoprotein receptor-related protein (LRP)5, Catenin beta (CTNNB)1, and Wingless-type MMTV integration site family, member (WNT)1 were also recorded [113]. Importantly, these lineage markers and transcriptomes were maintained from PDXs to the organoids even after the late generation [113]. 

Analysis of PC organoids showed that ERG was an effector of SPOP mutation in human and mouse PC models [114]. The analysis also revealed that BAF complexes were necessary for ERG-triggered basal-to-luminal transition, a typical feature of ERG activity in PC [115]. Mutation in the SPOP gene was shown to enhance the proliferation of PC cells by promoting ubiquitination and turnover of c-Myc oncoprotein as the organoid-forming capacity of SPOP-null murine prostate cells was more sensitive to c-Myc inhibition [116]. In another study, SPOP mutant organoids demonstrated a higher formation capability with more irregular borders and increased Ki-67 expression without AR upregulation compared with control ones [117]. Furthermore, it drove the in vivo tumorigenesis by regulation of PI3K/mTOR and AR signaling [117].

Loss of the chromatin remodeler chromodomain helicase DNA-binding protein 1 (CHD1) is a major genomic change found in human local and metastatic PC [118] and represents a distinct PC subtype characterized by SPOP mutation [119]. High levels of chromosomal rearrangements and higher genomic instability are associated with a potential defect in DNA damage repair [60,120]. Shenoy et al. revealed the role of CHD1 in PC development by using the CHD1-deleted mouse model and patient-derived organoid culture [121]. Collectively, these data suggest the efficiency of PC organoid models in studying and screening of genetic mutations in PC.

#### 3.3.2. PC Organoids for Predicting Anti-Cancer Drugs Sensitivity or Resistance

In PC, patient-derived organoids can provide preclinical models to perceive the precision medicine by drug screening of individual patient samples [24] (Figure 4). The ability to identify agents with high clinical effectiveness prior to clinical trials would enhance the efficiency of translating preclinical research into clinical practice.

Several established PC organoids from humans, mice, and dogs were successfully used to assess drug response. In humans, the seven established organoid lines by Gao et al., [112] were used for in vitro and in vivo screening of anti-cancer drugs. This 3D model was used for evaluation of the efficacy of three anti-cancer drugs, enzalutamide (the next-generation antiandrogen) and two phosphoinositide 3 kinase pathway inhibitors, everolimus, and BKM-120, which were used in clinical trials to treat a castration-resistant PC [112]. Data revealed that only one organoid line (MSK-PCa2, which harbors an AR amplification, PIK3R1 mutation, and PTEN loss) showed high sensitivity to enzalutamide, which supports the conclusion that molecular subtyping is necessary for targeting therapy [122]. Amazingly, this sensitive organoid line has demonstrated the same outcomes as a xenograft mouse model. 

In the same study, the organoid line established from PC circulating tumor cells instead of a core biopsy of metastatic sites is a distinct clinical possibility and is valuable as an efficacy-response indicator for overall survival in mCRPC [123]. The advantage of this model is to guide targeted therapy when access to tissue biopsy is impossible, as the case in bone marrow lesions of patients with metastatic PC.

In another study, Beshiri et al., found the sensitivity of PC organoids generated from PDXs of metastatic castrate-resistant PC to the PARP inhibitor, olaparib was similar to the responses recorded clinically in the patients [113]. 

The mutation in SPOP gene plays a role in the response of PC organoids to anti-cancer drugs. The PC organoids expressing SPOP W131R mutation are resistant to intrinsic BET inhibitors-induced cell growth arrest and apoptosis due to the stabilization of BET protein and activation of AKT-mTORC1 signaling [124,125,126]. To overcome this resistance, Yan et al. have investigated the anti-cancer effects of NEO2734 (novel BET-CBP/p300 dual inhibitor) in SPOP-mutated organoids from PC patients [127]. They demonstrated that NEO2734 showed the effectiveness in BET inhibitor (JQ1)-resistant SPOP mutant PC organoids [127]. 

In the mouse PC model, Chua et al., [128] assessed the sensitivity to anti-cancer drugs, using PC organoids generated from Nkx3.1^CreERT2/+^; Pten^flox/flox^; R26R–YFP/ + (NP) mice, which were previously used to screen therapeutic response in vivo [129]. Despite NP mice initially developed castration-sensitive PC, they later develop mCRPC that is responsive to dual treatment with an Akt inhibitor, MK-2206, and an mTOR inhibitor, MK-8669 (ridaforolimus) [129]. Using tamoxifen-induced NP mice, YFP-positive prostate cells were isolated for organoid culture and drug screening. The alone treatment of organoids with AR antagonist, enzalutamide or MK-8669 had minimal effects on organoid formation, while the combined treatment drastically inhibited organoid formation. Interestingly, these data are consistent with the known synergistic activities of AR and PI3K signaling in human PC [130].

In the dog PC model, the established organoids from urine cells also showed different sensitivities to the commonly used anti-cancer drugs [94,97]. The applied cell viability of these organoids revealed no response to a cyclooxygenase (COX)2 inhibitor, piroxicam. However, docetaxel and radiotherapy decreased the cell viability of these organoids in a dose-dependent manner.

#### 3.3.3. PC Organoids as An Effective Model to Study Mechanisms of Resistance to ADT

Resistance to anti-cancer drugs is one of the major clinical issues in cancer therapy. Treatment of metastatic PC is primarily directed at the androgen-signaling axis. Next-generation ADT (e.g., abiraterone and enzalutamide) have revealed great clinical success, however, almost all PC ultimately proceed towards an androgen-independent state. 

Recently, transcriptomic and genomic profiling of androgen-independent PC showed three general mechanisms of resistance in PC [131]. The first of which is activating mutations that lead to the restoration of androgen receptor signaling [132]. The second is a stimulation of bypass signaling to escape from androgen receptor blockade, as activation of the glucocorticoid receptor can compensate for the loss of androgen receptor signaling by activating a similar but distinguishable set of target genes necessary for the maintenance of the resistant phenotype [133]. The third is through the process of lineage plasticity, in which cancer cells get resistance by switching lineages from a drug target-dependent cell type to another none dependent one [134]. 

However, the molecular mechanisms that trigger resistance of PC to anti-cancer drugs are still poorly understood. Furthermore, the acquired resistance to ADT may drive to therapeutic vulnerabilities that can be tackled. Therefore, it is necessary for developing a model system to evaluate drug responses that simulate patient phenotypes and genotypes [131]. 

Since organoids can be genetically handled using CRISPR/Cas9 and shRNA systems, it can be a promising platform for elucidating the mechanisms of resistance to anti-cancer drugs [110]. Pappas et al., studied the effects of several anti-androgen molecules on PC murine organoids with different genotypes including Wild type (WT) genotype, PTEN loss, TP53 loss, or dual PTEN and TP53 loss [131]. They found that loss of TP53 did not cause resistance to the anti-androgenic molecules. However, loss of PTEN enhanced the resistance to the tested anti-androgens, as demonstrated previously [130]. Furthermore, they showed that dual loss of TP53 and PTEN resulted in complete resistance to the second-generation anti-androgens [131]. 

It is well known that patient-derived PC organoids are heterogeneous in phenotype and genotype [112,135]. Thus, responses to drugs greatly vary among human PC organoid lines. In the same study carried out by Pappas et al., a growth of MSKPCA2 organoids was substantially inhibited by anti-androgenic molecules. MSKPCA2 organoids expressed high levels of androgen receptor and luminal cell marker (CK8), while MSKPCA3 organoids showed resistance and expressed basal cell marker (CK5) [131].

The study carried out by Beshiri et al., [113] has determined the utility of the PDX-derived organoids as a model of AR-dependence, a cardinal feature of castration-resistant prostate adenocarcinoma. They analyzed the dynamics of AR and AR-responsive gene expression in organoids derived from highly (LuCaP 167) and moderately (LuCaP 77) castration-sensitive PDX models and from an AR-negative neuroendocrine control (LuCaP 145.2). Collectively, these data indicate the usefulness of PC organoids as a promising model to analyze the drug resistance in PC disease.

#### 3.3.4. PC Organoids to Study Cell of Origin of PC

Identifying cell of origin of PC might open new insights into the pathogenesis of PC initiation and may drive to better prognosis and selection of convenient therapy [119,136]. The cell of origin is the cell inside a normal tissue that subjects to oncogenic transformation to initiate tumor formation [137]. The cell origins of PC are still debated and differ among species and experimental models. The basal cells are considered as cells of origin, while luminal cells are commonly involved by genetic lineage-tracing in GEM models [138]. 

The 3D culture system of normal prostate organoid has been established and allows the growth and differentiation of both basal and luminal cell types [139]. This leads to establishing numerous models to study the cell of origin of PC [119]. 

In human PC, the luminal phenotype characterized by atypical glands and an absence of basal cells was observed [140], suggesting that PC originates from luminal cells. In contrast, it was reported that basal cells have the capacity for transformation in human PC [141,142] based on the expression of a number of stem cell markers, including α2β1 integrin, CD44, and CD133 [143,144]. 

Oncogenic insults of basal cells can trigger prostate tumorigenesis, suggesting a basal cell origin for mouse and human PC. For example, overexpression of ERG or activated Akt in Lin^−^Sca-1^+^CD49^hi^ basal cells induced prostatic intraepithelial neoplasia, and co-activation of Akt and AR signaling produced adenocarcinoma [145]. In another way, overexpression of Myc proto-oncogene protein and activated Akt in human basal cells gave rise to prostatic squamous adenocarcinoma [142]. 

The bioinformatic analysis revealed the enhanced aggressiveness of PC originated from luminal cells. This analysis could be valuable in the identification of patients who need urgent treatment and those who could benefit from active surveillance. The castration-resistant NKX3-1-expressing stem cells which reside in the luminal layer of an androgen-deprived prostate were shown to be the cell of origin in PC after PTEN deletion and give rise to both basal and luminal cells [146]. Also, organoids generated from CD38-low luminal cells expressing both basal (CK5 and p63) and luminal (CK8) cell markers showed higher organoid formation (4–5 times) and could regenerate into prostatic glands *in vivo* compared with CD38-high luminal cells, suggesting that it could be a useful biomarker to identify progenitor-like and inflammation-linked luminal cells that can launch human PC [147]. 

Park and colleagues [148] showed that both primary basal and luminal epithelial cells are cells of origin for PC using the organoid culture system. By overexpression of c-Myc and activated Akt in both cells, they found c-Myc-myrAkt1-transduced luminal xenografts showed well-differentiated acinar adenocarcinoma, whereas the basal xenografts were histologically more aggressive, suggesting that both kinds of cells could respond to the same oncogenic insults to initiate tumorigenesis, but the tumor phenotypes are different [148].

In PC mouse models, current evidence can support both a basal [149,150,151,152] and a luminal [128,146,153] CSC model of PC and can initiate PC with heterogeneous molecular signatures that are predictive of human patient outcomes. 3D culture of luminal cells derived from a PTEN-TP53-null mouse model was able to initiate adenocarcinoma or tumors with multilineage histological phenotypes [154]. 

In the dog PC model, we have established PC organoids from urine-derived CSCs. We showed that the basal cell-sorted organoids grew more efficiently than the luminal cell-sorted ones [94,97]. Moreover, the organoid-forming efficiency of basal cell-sorted organoids was significantly higher and had solid sphere structures compared with that of luminal cell-sorted organoids which had glandular structure. Interestingly, these data correspond with those recorded before in human prostate sorted cell-derived organoids [148], emphasizing the importance of this model in analyzing the cell of origin of PC in human. Compared with human, dogs are known to spontaneously develop PC [155] and have some similarities in the pathogenesis of the disease [156]. Therefore, dog PC may serve as a useful model for the diagnosis and treatment of PC in human.

## 4. Conclusions

Studies about PC have been restrained by the fact that current preclinical PC models do not accurately recapitulate the heterogeneity or complexity of the tissue microenvironment of PC. The advances in the development of organoids have thrilling possibilities for PC studies. The establishment of organoid biobanks and patient-derived organoids can facilitate the promotion of personalized medicine, for example, a high-throughput drug screening for therapeutic and toxic effects to identify the most effective and less toxic therapy for the individual patient. Additionally, an elucidation of the mechanism of drug resistance and a screening of genetic mutations responsible for the disease emergence and progression are now possible using this model. Future studies focusing on PC organoid culture will contribute to understanding the pathogenesis of human PC and identifying the new therapeutic target.

## Figures and Tables

**Figure 1 cancers-12-00777-f001:**
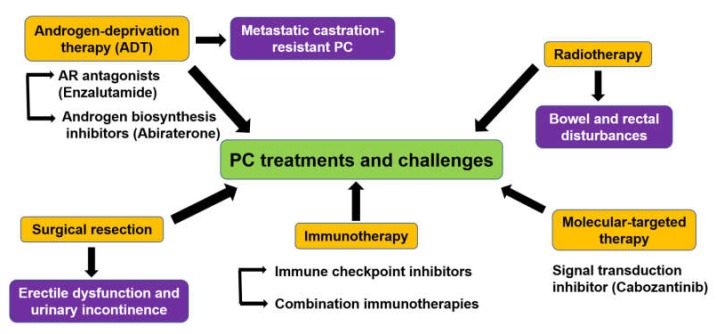
Outlines of current treatments of prostate cancer (PC) and their challenges. The recent treatments involve surgical resection, radiation therapy, chemotherapy, androgen-deprivation therapy (ADT), immunotherapy, and molecular-targeted therapy.

**Figure 2 cancers-12-00777-f002:**
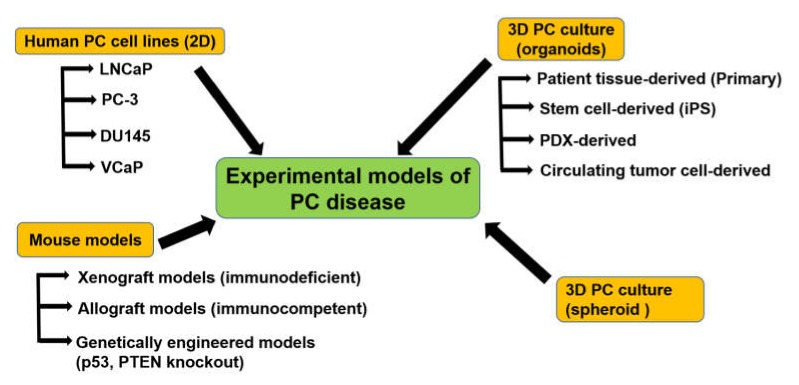
Main models to study PC disease and their subtypes. PDX: Patient-derived xenograft.

**Figure 3 cancers-12-00777-f003:**
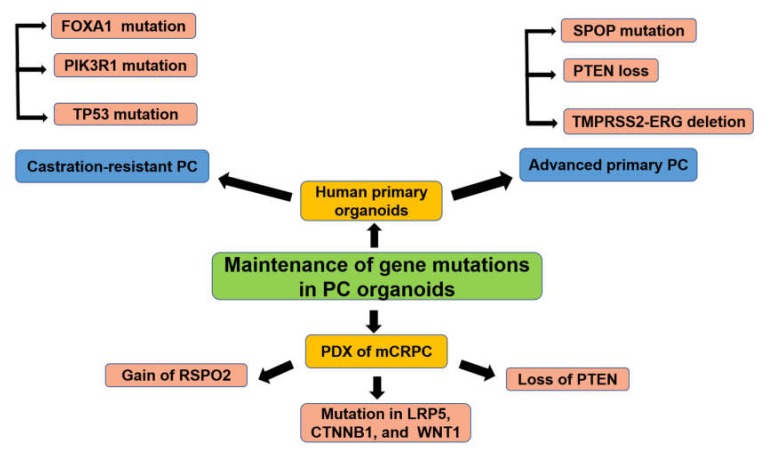
PC organoids for gene mutation screening. FOXA1: Forkhead box protein A1; PIK3R1: Phosphoinositide-3-kinase regulatory subunit 1; TP53: Total protein 53; SPOP: Speckle-type POZ protein; PTEN: Phosphatase and tensin homolog; TMPRESS2: Transmembrane protease, serine 2; ERG: Erythroblast transformation-specific-related gene; RSPO2: R-spondin 2; LRP5: Low-density lipoprotein receptor-related protein 5; CTNNB1: Catenin beta 1; WNT1: Wingless-type MMTV integration site family, member 1.

**Figure 4 cancers-12-00777-f004:**
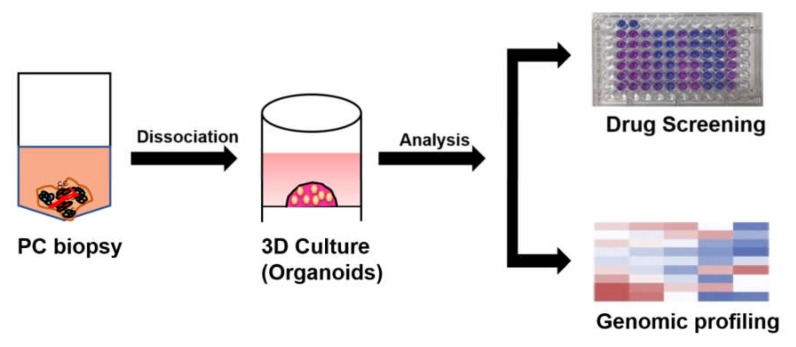
PC organoids for drug screening. After the biopsy samples are mechanically minced and enzymatically digested, the cell suspension is strained, washed and embedded within Matrigel to form organoids. After serial passaging, the 3D organoids can be used for drug screening as well as genomic profiling.

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
