# Peer review of "Development of Prostate Cancer Organoid Culture Models in Basic Medicine and Translational Research"

_cancers, 2020, doi:10.3390/cancers12040777_

Round 1
Reviewer 1 Report
In the present manuscript authors describe the application of PC organoids.
Introduction section should be improved: it is elusive and doesn’t refer to steroid receptors or other proteins that play a pivotal role in PC etiology and progression. It is necessary to better understand the basis of therapeutic strategies as Androgen deprivation therapy. Also the description of Prostate structure is not clear. The following reviews (doi: 10.18632/oncotarget.6220; doi: 10.3390/cancers11101418; 10.3389/fonc.2018.00002) could give useful cues.
The grouping of PC cell lines on the basis of their characteristics (AR positive; ER positive, metastatic and etc) is useful to the readers. Please include in table 1 the ERa or ERb positivity.
Figure 2 shows spheroids and organoids. Please explain the main differences among them.
Please add a legend to figure 4.
Author Response
Responses to Reviewer: 1
Q1. Introduction section should be improved: it is elusive and doesn’t refer to steroid receptors or other proteins that play a pivotal role in PC etiology and progression. It is necessary to better understand the basis of therapeutic strategies as Androgen deprivation therapy. Also the description of Prostate structure is not clear. The following reviews (doi: 10.18632/oncotarget.6220; doi: 10.3390/cancers11101418; 10.3389/fonc.2018.00002) could give useful cues.
A1. Thanks for valuable comments. We referred to the steroid receptors that play a pivotal role in PC etiology and progression. Also, we modified the parts related to the structure of prostate and cited the papers which the reviewer suggested (page 2, line 42-67; page 3, line 95).
Q2. The grouping of PC cell lines on the basis of their characteristics (AR positive; ER positive, metastatic and etc) is useful to the readers. Please include in table 1 the ERa or ERb positivity.
A2. Following the suggestion, we added the information on the expression of ER a and b for the cell lines (page 5, new Table. 1).
Q3. Figure 2 shows spheroids and organoids. Please explain the main differences among them.
A3. In general, the spheroids are developed from cancer cell lines or tumor specimens as freely floating cell clusters in ultra-low attachment plates, whereas the organoids are generated from stem cells in primary tissues. Organoids are more complex and exhibit more in vivo-like features than spheroids.
Q4. Please add a legend to figure 4.
A4. Following the suggestions, we added a legend to Figure 4 (page 9, line 296-299).
Reviewer 2 Report
This review manuscript provided a detailed description of PC organoid culture models along with their impact on gene mutations, anti-cancer drug sensitivity, COO, and ADT resistance. In terms of gene mutation screening, the researchers discussed about some crucial gene mutations of PCa namely, FOXA1, TP53, PTEN, WNT1, and so on. Then they also displayed the sensitivity of mutated cells to different novel agents that included enzalutamide (the next-generation antiandrogen), phosphoinositide 3 kinase pathway inhibitors (everolimus and BKM-120), Akt inhibitor-258 MK-2206, mTOR inhibitor-MK-8669 (ridaforolimus), BET-CBP/p300 dual inhibitor (NEO2734). In regard to COO, they cited the work showing that both primary basal and luminal epithelial cells are COO for PCa using the organoid culture system. Indeed, acinar adenocarcinoma was associated with transduced luminal xenografts and more aggressive nature was evident in basal xenografts. They also mentioned that loss of PTEN resulted in resistance to the anti-androgen therapy and that dual loss of TP53 and PTEN was associated with resistance to the second-generation anti-androgens.
The manuscript was very comprehensive that also denoted other models with their potential limitations. These models included xenograft model like patient derived xenograft in immunodeficient mice. The examples of mice include NOG (NOD/Scid/IL2Rγnull), NSG (NOD/Scid/IL2Rγnull), and NOJ 138 (NOD/Scid/Jak3null) mice. The manuscript also provided with PSA and AR profiles of different PCa cell lines that is crucial in this field of research.
Overall, the manuscript is nicely written, contains relevant references, and supports evidence for organoid models to be used in translational research.
Author Response
Responses to Reviewer: 2
Thanks for reviewing our paper and give us a valuable comment. Following the comment of Review 1, we added the information.
Responses to Reviewer: 1
Q1. Introduction section should be improved: it is elusive and doesn’t refer to steroid receptors or other proteins that play a pivotal role in PC etiology and progression. It is necessary to better understand the basis of therapeutic strategies as Androgen deprivation therapy. Also the description of Prostate structure is not clear. The following reviews (doi: 10.18632/oncotarget.6220; doi: 10.3390/cancers11101418; 10.3389/fonc.2018.00002) could give useful cues.
A1. Thanks for valuable comments. We referred to the steroid receptors that play a pivotal role in PC etiology and progression. Also, we modified the parts related to the structure of prostate and cited the papers which the reviewer suggested (page 2, line 42-67; page 3, line 95).
Q2. The grouping of PC cell lines on the basis of their characteristics (AR positive; ER positive, metastatic and etc) is useful to the readers. Please include in table 1 the ERa or ERb positivity.
A2. Following the suggestion, we added the information on the expression of ER a and b for the cell lines (page 5, new Table. 1).
Q3. Figure 2 shows spheroids and organoids. Please explain the main differences among them.
A3. In general, the spheroids are developed from cancer cell lines or tumor specimens as freely floating cell clusters in ultra-low attachment plates, whereas the organoids are generated from stem cells in primary tissues. Organoids are more complex and exhibit more in vivo-like features than spheroids.
Q4. Please add a legend to figure 4.
A4. Following the suggestions, we added a legend to Figure 4 (page 9, line 296-299).

Round 2
Reviewer 1 Report
Authors improved the manuscript according to my suggestions. I approved the manuscript in the present form.